# Root Production and Microbe-Derived Carbon Inputs Jointly Drive Rapid Soil Carbon Accumulation at the Early Stages of Forest Succession

**Ruiqiang Liu [1], Yanghui He [1], Zhenggang Du [1], Guiyao Zhou [2], Lingyan Zhou [2], Xinxin Wang [2], Nan Li [2], Enrong Yan [2], Xiaojuan Feng [3], Chao Liang [4] and Xuhui Zhou [1,2,***

[1] Northeast Asia Ecosystem Carbon Sink Research Center (NACC), Center for Ecological Research, Key Laboratory of Sustainable Forest Ecosystem Management-Ministry of Education, School of Forestry, Northeast Forestry University, Harbin 150040, China

[2] Center for Global Change and Ecological Forecasting, Tiantong National Field Station for Forest Ecosystem Research, Shanghai Key Lab for Urban Ecological Processes and Eco-Restoration, School of Ecological and Environmental Sciences, East China Normal University, Shanghai 200062, China

[3] State Key Laboratory of Vegetation and Environmental Change, Institute of Botany, Chinese Academy of Sciences, Beijing 100093, China

[4] Key Laboratory of Forest Ecology and Management, Institute of Applied Ecology, Chinese Academy of Sciences, Shenyang 110016, China

*   Correspondence: xhzhou@nefu.edu.cn

**Abstract:** Plants and microbes are the primary drivers in affecting the formation and accrual of soil organic carbon (SOC) for natural ecosystems. However, experimental evidence elucidating their underlying mechanisms for SOC accumulation remains elusive. Here, we quantified plant and microbial contributions to SOC accrual in successional subtropical forests by measuring leaf-, root-, and microbial biomarkers, root and leaf litter inputs, and microbial C decomposition. The long-term monitoring results showed that SOC accumulated rapidly at the early-successional stage, but changed little at the mid- and late-successional stages. SOC accrual rate was positively correlated with fine-root production and microbial C turnover, but negatively with annual litterfall. Biomarker data exhibited that the rapid SOC accumulation was jointly driven by root- and microbe-derived C inputs from the early- to mid-successional stages. In contrast, aboveground litterfall considerably contributed to soil C accrual from the mid- to late-successional stages compared to belowground processes, although SOC accumulation is low. Our study revealed the importance of root production and microbial anabolism in SOC accrual at the early stages of forest succession. Incorporating these effects of belowground C inputs on SOC formation and accumulation into earth system models might improve model performance and projection of long-term soil C dynamics.

**Keywords:** biomarker; carbon inputs; microbial anabolism; microbial carbon turnover; root functional traits; root production



## 1. Introduction

Forests cover approximately 21.9% of terrestrial land surface, making them the largest reservoir of soil organic carbon (SOC) across all terrestrial ecosystems [1,2]. Global forests sequester approximately 2.4 Pg C yr$^{-1}$ from the atmosphere, offsetting approximately 24% of the $CO_2$ releases from fossil fuel combustion [3,4]. This large quantity of sequestrated C is primarily controlled by the balance between plant-derived inputs via vegetation productivity and C consumption from microbial metabolic activities [5]. Importantly, rapid climatic change and human activities have considerably altered the soil C balance, leading to potential changes in soils from C sinks to C sources [6]. Therefore, understanding plant and microbial controls on soil C sequestration is crucial in developing sustainable strategies of forest management for climate-change mitigation and adaptation.

Over several decades, soil C inputs from aboveground litter were thought to be the dominant factors influencing soil C sequestration [7,8]. However, this view has been challenged by several studies, suggesting that SOC sequestration is primarily driven by belowground root litter inputs and microbial activity [9–11]. Root litter is typically more lignified and chemically recalcitrant than aboveground litter [12]. In addition, root labile compounds are more likely to be bound to soil metallic ions and mineral particles that help stabilize soil aggregates [13]. With greater chemical and physical protection, root litter decomposes more slowly in soils with 1.3–2.4 times the amount of C residence time relative to that of shoots [12]. Moreover, roots usually associate with mycorrhizal fungi to guide 3%–30% of photosynthetic C from trees to soils [14]. The cell walls of these fungi are composed primarily of chitin, which is recalcitrant to decomposition, and the lifespan of external hyphae averages only five to six days [15]. The great production of live hyphae allows the accrual of mycelial residues. As such, mycorrhizal fungi are a key conduit for plant-derived C inputs to soil, and a primary determinant of soil C sequestration in many ecosystems [16–18].

In addition to C inputs from plants and mycorrhizae, soil C sequestration is also controlled by microbial decomposition and anabolism [19,20]. Soil microorganisms convert organic matter into microbial necromass (represented by amino sugar content) during the decomposition of plant materials. Although microbial biomass constitutes a tiny fraction of ecosystem SOC, fast turnover within microbial communities often results in a massive accrual of microbial necromass [21]. These microbial products, containing both labile and recalcitrant compounds, are often bound with minerals that can persist in soils, and thus contribute to a stable C pool [20]. As a result, microbial necromass residues often comprise the bulk of the SOC pool [21]. Although the contributions of microbial necromass and root residues to the SOC pool have been studied, the knowledge about how environmental variables and microbial activities mediate the size of microbial- and root-derived C pools remains limited [9,20,22]. Less is known about the relative retention of C from leaf, root and microbial necromass residues in forest soils.

Forest succession after disturbance is a naturally occurring process of plant community replacement, which has highly important implications for ecological restoration, plant diversity, and ecosystem functioning as well as climate change mitigation [23–25]. Natural succession usually facilitates an increase in SOC, especially in tropical regions [26,27]. Accumulation rates of SOC are known to vary across the stages of forest succession [28–30]. Although root and microbial controls on soil C accrual have been widely investigated [9,31], the main driver of root- and microbially-derived C accumulation remains elusive. This knowledge gap largely impedes accurate predictions regarding long-term SOC dynamics in forest ecosystems.

In this study, we took advantage of three successional subtropical forests in Eastern China to explore the effects of root- and microbially-derived C inputs on soil C accrual. We quantified the SOC accrual rate at different successional stages based on long-term in situ measurements (from 2004 to 2018). Aboveground litterfall, fine-root production, root morphological and nutrient traits, and soil microbial C turnover were measured to explore their relationships with SOC accrual rate. Three groups of widely-accepted biomarkers (i.e., cutin, suberin, and amino sugar) [32,33] were used to quantify the relative contributions of aboveground litter-, root-, and microbial necromass-C to SOC accumulation. We hypothesized that: (i) the contributions of root- and microbial necromass-C to the SOC pool increased with forest succession; and (ii) root production and microbial C turnover were greater at the early-than late-successional stages, resulting in a faster SOC accumulation rate from early- to mid-successional forests.

## 2. Methods

### 2.1. Experimental Site and Forest Successional Stages

The experimental site was located within the Tiantong National Forest Ecosystem Observation and Research Station (29°48′ N, 121°47′ E) in Zhejiang province, China. Mean an-

nual temperature in this area is 16.2 °C and average annual precipitation is 1374.7 mm [34]. Soils are classified as hilly red and yellow earths (Acrisols and Cambisols in the FAO soil classification, respectively), and pH ranges between 4.5–5.1 [34,35]. Secondary forests are managed through repeated cutting. According to the differences of abandoned ages, these forests are representative of different successional stands and form a successional chronosequence [34]. Experimental blocks were established within secondary successional forests, which were composed of three representative successional stages (early stage: 25 years young mixed woody community, middle stage: 55 years sub-climax *Schima superba* community, and late stage: 120 years climax *Castanopsis fargesii* Franch. community) [25,26]. The slope and elevation of the three successional forest stands were similar to one another, and there was a minimum 100 m buffer from the forest edge around each stand [34]. The distances among three successional forest stands were 1000–3000 m. Within each of the three stands, six 10 × 10 m plots were established. Tree height, basal area, stem density, and community composition were measured in August, 2015. Details of sampling sites, including the map of the study area and edaphic conditions community characteristics were described by Liu et al. (2019) [25], community characteristics are shown in Table S1 of the Supplementary Materials.

### 2.2. Measurements of SOC Accrual Rate and Biomarkers

Soils were sampled using PVC cylinders with a diameter of 10 cm and length of 20 cm. PVC cylinders were hammered into the soil and then dug out with a spade. Five soil samples were taken per plot during the growing season (August), 2004, 2014, and 2018. Stones and plant residue, including roots, were removed from soil samples using tweezers. The five fresh soil samples per plot were mixed to obtain one composite sample. Composite soil samples were air dried at room temperature. Next, 50 g of air-dried soil were taken from each composite sample per plot and were sieved to 0.149 mm. Soil organic carbon (SOC) was then measured using the Walkley–Black wet digestion method [34]. SOC accrual rates at different successional stages were then calculated as the difference in SOC concentrations between 2004 and 2018 divided by the sampling interval time of 14 years. Soil bulk density was measured using stainless steel rings with 100 $cm^3$ volume ($n = 5$) from each plot from 0 to 20 cm depths in all sampling years. SOC stock (g $m^{-2}$) at the depth of 0–20 cm was calculated as following:

$$\text{SOC stock} = D \times BD \times OC \times 20 \tag{1}$$

where D is soil depth (cm), BD is bulk density (g $cm^{-3}$), and OC is the soil organic carbon concentration (g $kg^{-1}$).

Plant-derived biomarkers, including cutin and suberin compounds, were measured following the procedures outlined by Feng et al. (2008) [36]. Briefly, ten grams of fresh soil samples were freeze-dried and extracted using 30 mL of methanol, dichloromethane: methanol (1:1; *v/v*), and dichloromethane, respectively. The three solvent chemical extracts were combined and then filtered through a glass fiber filter (Whatman GF/A and GF/F). Liquid samples were concentrated via rotary evaporation and were completely dried using nitrogen gas. The solvent-extracted soil sub-samples (2 g) were then subjected to base hydrolysis to release ester-bound lipids. The air-dried soil residues were heated at 100 °C for 3 h in teflon-lined bombs with 1 M methanolic KOH (20 mL). The extracted liquids were then acidified to pH 1 with 6 M HCl, and solvents were removed via rotary evaporation. Lipids were recovered from the water phase by liquid–liquid extraction with diethyl ether, concentrated by rotary evaporation, and dried under nitrogen gas in 2 mL glass vials. The composition and concentration of extractable compounds were then analyzed using gas chromatography/mass spectrometry (GC/MS).

$$\text{Suberin concentration } (\Sigma S) = C20 - C32 \, \omega\text{-hydroxyalkanoic acids} + C20 - C32 \, \alpha,\omega\text{-dioic acids} + C18 \, 9,10\text{-epoxy-dioic acid} \tag{2}$$

$$\text{Cutin concentration } (\Sigma C) = \text{C14, C15, C17 mid-chain hydroxyalkanoic acids + C16 mono-} \atop \text{and dihydroxyalkanoic acids and dioic acids} \tag{3}$$

$$\text{Unidentified components (originated from suberin or cutin, } \Sigma S \vee C) = \text{C16, C18 } \omega\text{-hydroxyalkanoic acids} \atop + \text{ C18 di- and trihydroxy acids + C18 9,10-epoxy-dioic acid + C16, C18 } \alpha,\omega\text{-dioic acids} \tag{4}$$

$$\text{Sum of suberin and cutin } \left( \sum SC \right) = \sum S + \sum C + \sum S \vee C \tag{5}$$

$$\text{Relative contribution of root- vs. leaf-derived C to SOC pool} = \left( \sum S + \sum S \vee C \right) / \left( \sum C + \sum S \vee C \right) \tag{6}$$

Soil amino sugars were measured by the classical protocol according to Zhang and Amelung (1996) [37]. Ten grams of air-dried soil were ground to pass through a 0.25 mm mesh sieve and hydrolyzed with 6 M HCl for 8 h at 105 °C. The supernatant was filtered and adjusted to pH 6.6–6.8 and passed through a centrifuge. The extracts were then freeze-dried and re-dissolved in methanol. The recovered amino sugars were transformed into aldononitrile derivatives and analyzed with an Agilent 6890A GC equipped with a flame ionization detector (FID), using a capillary column (HP-5). Myoinositol was used as an internal standard, which was added prior to purification, to quantify soil amino sugar. The recovery efficiency of the amino sugars was measured using methylglucamine as a recovery standard before derivatization. Three kinds of amino sugars were analyzed by Agilent 6890A GC, including glucosamine (GluN), galactosamine (GalN), and Muramic acid (MurA). GluN was derived from fungal and bacterial cell wall, while MurA was exclusively originated from the bacterial cell wall [38]. Because the origin of GalN has not been identified, it was involved in estimating microbial necromass C [10]. Total microbial necromass C concentrations were calculated by summing the amounts of fungal and bacterial necromass C.

$$\text{Fungal necromass C} = (\text{GluN}/179.17 - 2 \times \text{MurA}/251.23) \times 179.17 \times 9 \tag{7}$$

$$\text{Bacterial necromass C} = \text{MurA} \times 45 \tag{8}$$

where 179.17 and 251.23 represent the molecular weight of GluN and MurA, respectively, and 9 and 45 are the conversion coefficients of GluN to fungal necromass C and MurA to bacterial necromass C, respectively [38].

### 2.3. Measurements of Annual Fine-Root Production and Litterfall

Fine-root biomass was measured within 24 h after soil samples were collected in August, 2004 and 2018. Soil cores were left intact to preserve root structure prior to all analyses. In the laboratory, soil cores were gently broken apart, and intact fine roots were carefully picked out using tweezers. Root samples were oven-dried at 65 °C for 48 h and weighed. Fine-root production was calculated as fine-root biomass × turnover rate [39]. The root turnover rate was measured using litter-bag method described by Shi et al. (2005) [40].

To measure annual litterfall, we placed two 0.7 × 0.7 m collection boxes at each 10 × 10 m plot across successional forests. Litterfall was collected monthly in 2004, 2013, and 2018. Litter samples were weighed after drying at 65 °C for 48 h.

### 2.4. Measurements of Fine-Root Morphology and N Concentration

Fine-root (diameter < 2 mm) morphological traits, including root length density, specific root length, root tissue density and root diameter were measured within 24 h after root samples were collected in August, 2018. Roots were washed to remove the remaining soil and organic matter particles, and stored in water to prevent them from drying before morphological analysis [41,42]. Five grams of fresh fine roots were used to investigate the proportions of fibrous and pioneer roots. Fibrous roots were defined as first and second order roots with diameter < 1 mm, and the remaining fine roots were deemed pioneer roots [43]. Then, 2 g of fresh mass from the fine-root sub-samples were scanned at

a resolution of 400 dpi using a desktop scanner (Epson Expression 10,000 XL scanner) to measure root morphology. Background impurities (e.g., shadows and spots) in scanned images were manually erased using Adobe Photoshop version 8.0 LE (Adobe Systems). WinRHIZO Arabidopsis version 2012b (Regents Instruments Inc., Quebec, QC, Canada) were used to analyze the scanned images to determine fine-root length, mean diameter, and root volume. Fine-root length density (RLD) was defined as the total root length per unit of soil volume. Specific root length (SRL) was calculated as the ratio between total root length and root dry mass. Root tissue density was defined as root dry mas per unit of root volume. Root samples were oven-dried and ground in a Wiley mill to pass a screen mesh with 0.149 mm hole size. Fine-root N concentration was then measured using a Vario MAX elemental analyzer (Elementar, Germany).

### 2.5. Measurements of Soil Respiration and Soil Carbon Residence Time

Soil respiration ($Rs$) and its components were measured from January to December, 2018. Four $0.7 \times 0.7$ m sub-plots were established within each of the $10 \times 10$ m plots and intentionally positioned to exclude existing trees in August 2016. Two subplots within each plot were then chosen randomly as either a control group, the other two were set as a trenched treatment group. Trenches were dug to a depth of 0.8 m (with little fine root distribution below this depth) with a width of 40 cm, and lined with nylon mesh to exclude living roots. All trenches were immediately back-filled with the excavated soils to minimize the disturbance of trenching. In total, there were 24 sub-plots (4 sub-plots $\times$ 6 plot replicates) established within each successional stand. A PVC collar (diameter: 20 cm, height: 10 cm) was then placed 5 cm below ground in each sub-plot. $Rs$ and $R_H$ were measured once a month between 8:00 and 12:00 AM from December 2016 to December 2018, using a LI-COR 8100 portable soil $CO_2$ flux system (LI-COR. Inc., Lincoln, NE, USA). Soil $CO_2$ flux and its components were calculated from January to December 2018. Heterotrophic respiration ($R_H$) was defined as $CO_2$ efflux in the trenched sub-plots. Root respiration ($R_{root}$) was calculated as the difference in $CO_2$ efflux between control ($Rs$) and trenched ($R_H$) treatments. Specific root respiration was calculated as the ratio of annual $R_{root}$ to root biomass (0–20 cm soil layer).

Soil microbial biomass C (MBC) was measured in February, May, August, and December 2018, using the chloroform fumigation extraction technique [29]. Mineral soils were sampled at a depth of 20 cm at five points per plot at each successional stage. Fresh soil samples were then sieved through a 2 mm sieve. Five grams of fumigated and non-fumigated fresh soil sub-samples from each plot were prepared according to Brookes et al. (1985) [44]. Samples were mixed with 20 mL $K_2SO_4$ (0.5 M) and shaken for 30 min. The mixture was then centrifuged at 4000 rpm for 10 min and filtered through Whatman 42 paper and a 0.45 μm filter membrane. Organic C concentrations in the $K_2SO_4$ extracts were then determined using a SHIMADZU TOC-VCPH/CPN analyzer (Germany). Soil MBC concentration was calculated by subtracting total dissolved organic C (DOC) of non-fumigated sub-samples from that of the fumigated sub-samples with a conversion factor of 0.45 [44].

Specific microbial respiration (referred to as microbial C turnover), which was calculated as the ratio of annual $R_H$ to MBC stock. Soil C residence time was calculated as the ratio of SOC stock to annual $R_H$.

### 2.6. Data Analysis

The impacts of forest succession on SOC accrual rate, soil cutin and suberin concentrations, $Rs$, $R_H$ and $R_{root}$, annual litterfall, fine-root production and functional traits, specific microbial respiration, microbial biomass C concentration, microbial necromass C concentration, and C residence time were tested using a one-way ANOVA. Correlations among SOC accumulation rate, SOC components, annual litterfall, microbial properties, and fine-root functional traits were quantified with Pearson correlation analysis in R (R Core Team, 2014). Statistical analyses were conducted in SPSS 16.0 for Windows (SPSS. Inc.,

Chicago, IL, USA), and figures were drawn with Sigmaplot 10.0 (Systat Software Inc., San Jose, CA, USA) and *R* software (Systat Software Inc., San Jose, CA, USA).

## 3. Results

### 3.1. Soil Organic Carbon (SOC) Accumulation

Soil organic carbon concentration at the soil layer of 0–20 cm in early-successional forests was 2.8% in 2004, 3.0% in 2013, and 3.9% in 2018, with an average accrual rate of 0.079% $yr^{-1}$ (Figure 1a, $p < 0.05$). Soil organic carbon stock in early-successional forests was 71.6 t $ha^{-1}$ in 2004, 76.6 t $ha^{-1}$ in 2013, and 95.2 t $ha^{-1}$ in 2018, with an average accrual rate of 1.7 t C $ha^{-1}$ $yr^{-1}$ (Figure 1b). In contrast, in mid- and late-successional forests, SOC concentrations and stocks did not significantly change over time. As expected, SOC concentration and stock were greater in late-successional forests compared to those in early- and mid- successional ones (Figure 1, $p < 0.05$).

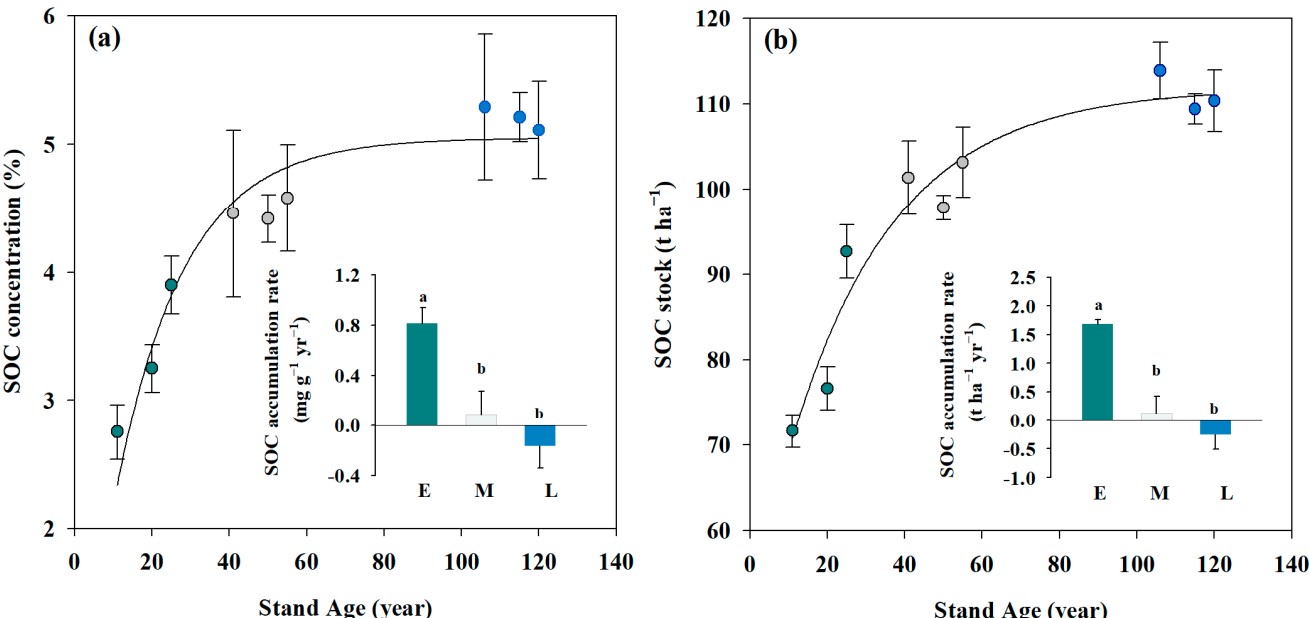

**Figure 1.** Long-term changes in SOC concentration (panel (**a**)) and stock (panel (**b**)) in the top 20 cm of soils at three successional stages. Soils were sampled in the years: 2004, 2014, and 2018. Dark cyan dots: measurements in the early-successional forests, gray dots: measurements in the mid-successional forests, and blue dots: measurements in the late-successional forests. The insets in panel (**a**,**b**) show the SOC accrual rate at each stage of forest succession, where E = early-successional stage, M = mid-successional stage, and L = late-successional stage. The lines shown in (**a**,**b**) were fitted to the data. Different lowercase letters indicate significant differences among successional stages, $p < 0.05$. Dots and bar charts in the figures represent mean values, vertical bars represent the standard error.

### 3.2. Changes in Soil Cutin, Suberin, and Amino Sugar along the Forest Succession

Soil cutin, suberin, and amino sugar concentrations at the soil layer of 0–20 cm varied significantly along the forest succession (Figure 2, $p < 0.05$). Soil cutin concentration was lower in early- (221.6 mg $kg^{-1}$ soil) and mid- (201.0 mg $kg^{-1}$ soil) successional forests compared to that in late-successional ones (477.1 mg $kg^{-1}$ soil; Figure 2a, $p < 0.05$). Soil suberin concentration ranged from 387.6 to 406.7 mg $kg^{-1}$ across the successional stages with the lowest value in early-successional forests (Figure 2a, $p < 0.05$). Relative contribution of root- vs. leaf-derived C to the SOC pool (indicated by the ratio of suberin to cutin in soils) decreased with the forest succession (Figure 2a, $p < 0.05$). The contribution of root-derived C to the SOC pool was greater than that of leaf-derived C at the early- and mid-successional stages, but lower at the late-successional stage (Figure 2a, $p < 0.05$).

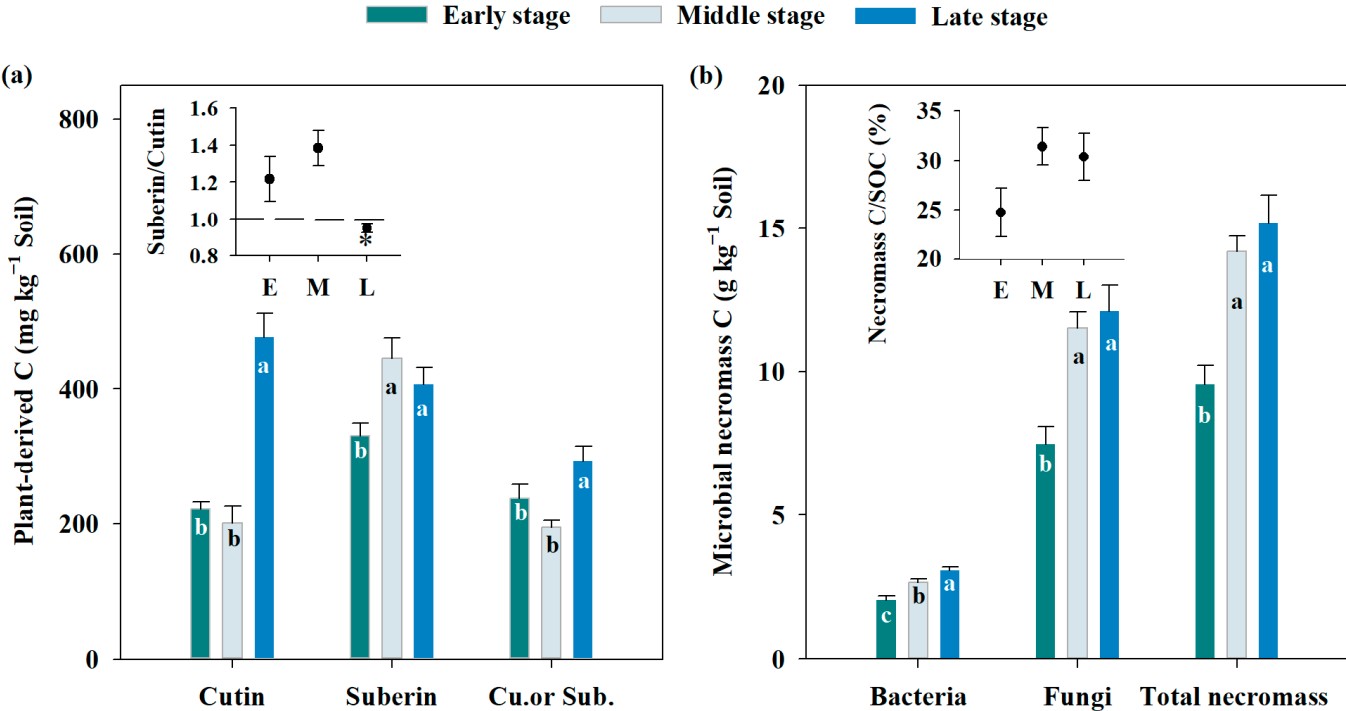

**Figure 2.** Cutin and suberin (panel (**a**)) and microbial necromass C concentrations (panel (**b**)) in the top 20 cm of soils at three successional stages. Note: E = early-successional stage, M = mid-successional stage, and L = late-successional stage. Suberin/Cutin = ($\sum$Suberin + $\sum$Cu. or Sub.)/($\sum$Cutin + $\sum$Cu. or Sub.). Cu. or Sub. represents the unidentified components originated from suberin or cutin. Different lowercase letters indicate significant differences among successional stages, $p < 0.05$. The asterisks (*) in the insets also signify significant differences, $p < 0.05$. Dots and bar charts represent mean values, vertical bars represent the standard error.

Soil microbial necromass C concentration (represented by amino sugar), including soil fungal and bacterial variations, significantly increased along with the forest succession (Figure 2b, $p < 0.05$). Microbial necromass C concentration at the early-successional stage was 9.6 g kg$^{-1}$, which was lower than those at the mid- and late-successional stages (14.2 and 15.2 g kg$^{-1}$, respectively). Similarly, soil fungal necromass C concentration was larger at the mid- than early-successional stage (54%), but exhibited no significant difference between the mid- and late-successional ones. Soil bacterial necromass C concentration was also larger at the mid- than early -successional stages (30%) and at the late- than mid-successional one (14%). Across forest succession, microbial necromass C accounted for 25%–30% of SOC. However, the contribution of microbial necromass C to the SOC pool did not significantly change along with the forest succession (inset in Figure 2b, $p > 0.1$).

*3.3. Annual Litterfall, Fine-Root Production, and Functional Traits*

Across the sampling years, annual fine-root production and biomass decreased along with the forest succession (Figures 3a and 4e, $p < 0.05$). In contrast, annual litterfall was greatest at the late-successional stage, intermediate for the mid-successional stage, and lowest at the early-successional stage (Figure 3b, $p < 0.05$). SOC accrual rate was positively correlated with annual fine-root production, but negatively with annual litterfall (Figure 3c,d, $p < 0.01$). Forest succession also changed fine-root functional traits. Specific root length (SRL) and root N concentration increased along the forest succession (Figure 4c,d). Root length density (RLD), root diameter, root tissue density (RTD), and fibrous/pioneer root ratio did not change along the forest succession (Figure 4b,d,f,g; $p > 0.1$). Root respiration was 2.27 μmol m$^{-2}$ s$^{-1}$ on average based on a one-year measurement (in 2018) at the early-successional stage, which was significantly greater than those at the

mid- (1.51 µmol m$^{-2}$ s$^{-1}$) and late-successional stages (1.41 µmol m$^{-2}$ s$^{-1}$, Figure 5a–c, $p = 0.03$). Specific root respiration (referred to as root respiration per unit root biomass) was 0.22 mg $CO_2$-C g$^{-1}$ dry root h$^{-1}$ at the late-successional stage, which was significantly greater than those at the early- (0.15 mg $CO_2$-C g$^{-1}$ dry root h$^{-1}$) and mid-successional stages (0.17 mg $CO_2$-C g$^{-1}$ dry root h$^{-1}$, Figure 5f, $p = 0.01$). SOC accrual rate was tightly correlated with specific root respiration, root biomass, SRL, and fibrous/pioneer root ratio (Figure 6, $p < 0.05$). There also appeared to be tight correlations among specific root respiration, root biomass, and SRL (Figure 6, $p < 0.05$).

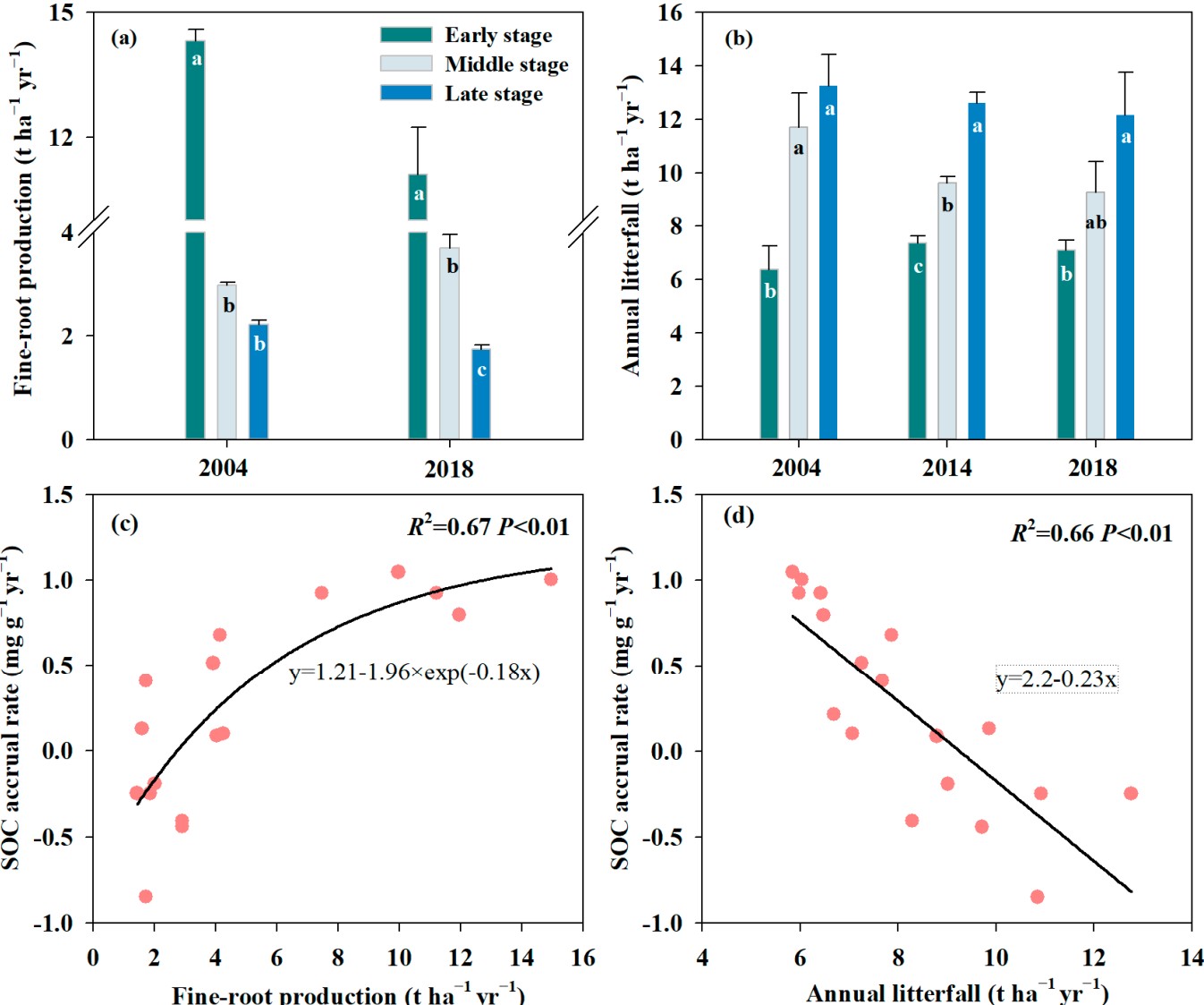

**Figure 3.** Long-term changes in fine-root production (panel (**a**)) and litterfall biomass (panel (**b**)) across forest succession, as well as their relationship with SOC accumulation rates. Fine-root production (panel (**c**)) and litterfall biomass (panel (**d**)) were measured in 2018. Different lowercase letters indicate significant differences among successional stages, $p < 0.05$. Dots and bar charts represent mean values, vertical bars represent the standard error.

### 3.4. Soil C Residence Time and Specific Microbial Respiration

Based on the monthly measurement in 2018, microbial respiration ($R_H$) was 2.51, 2.05, and 2.12 $\mu$mol $CO_2$ m$^{-2}$ s$^{-1}$ on average at the early, middle, and late stages of forest succession, respectively, with no significant difference among the three stages (Figure 5a–c, $p > 0.05$). Soil C residence time, which was calculated as the ratio of SOC stock to annual $R_H$, was 397 years at the early-successional stage, which was lower than that at the late one (613 years, Figure 5e, $p < 0.05$). Soil microbial C biomass (MBC) was 792 mg kg$^{-1}$ on average at the late-successional stage based on the seasonal measurement, which was greater than that at the early-successional stage (689 mg kg$^{-1}$, Figure 5d, $p = 0.07$). Specific microbial respiration (i.e., the ratio of annual $R_H$ to MBC stock) was 12.1 mg $CO_2$-C g$^{-1}$ MBC h$^{-1}$ at the early-successional stage, which was significantly greater than that at the mid- (9.35 mg $CO_2$-C g$^{-1}$ MBC h$^{-1}$) and late-successional stages (9.05 mg $CO_2$-C g$^{-1}$ MBC h$^{-1}$; Figure 5f, $p < 0.05$). Specific microbial respiration was positively correlated with the SOC accumulation rate (Figure 6, $p < 0.05$).

Based on the biomarker and soil C inputs/outputs data, we developed a conceptual framework to explain plant and microbial control on SOC formation and accrual in subtropical forests (Figure 7). We found that high root production and fast microbial C turnover (referred to as specific microbial respiration) resulted in rapid SOC accrual in early-successional forests. Greater aboveground C inputs led to higher leaf-derived C accrual but had little effect on the accrual of total soil C stock in the late-successional forests.

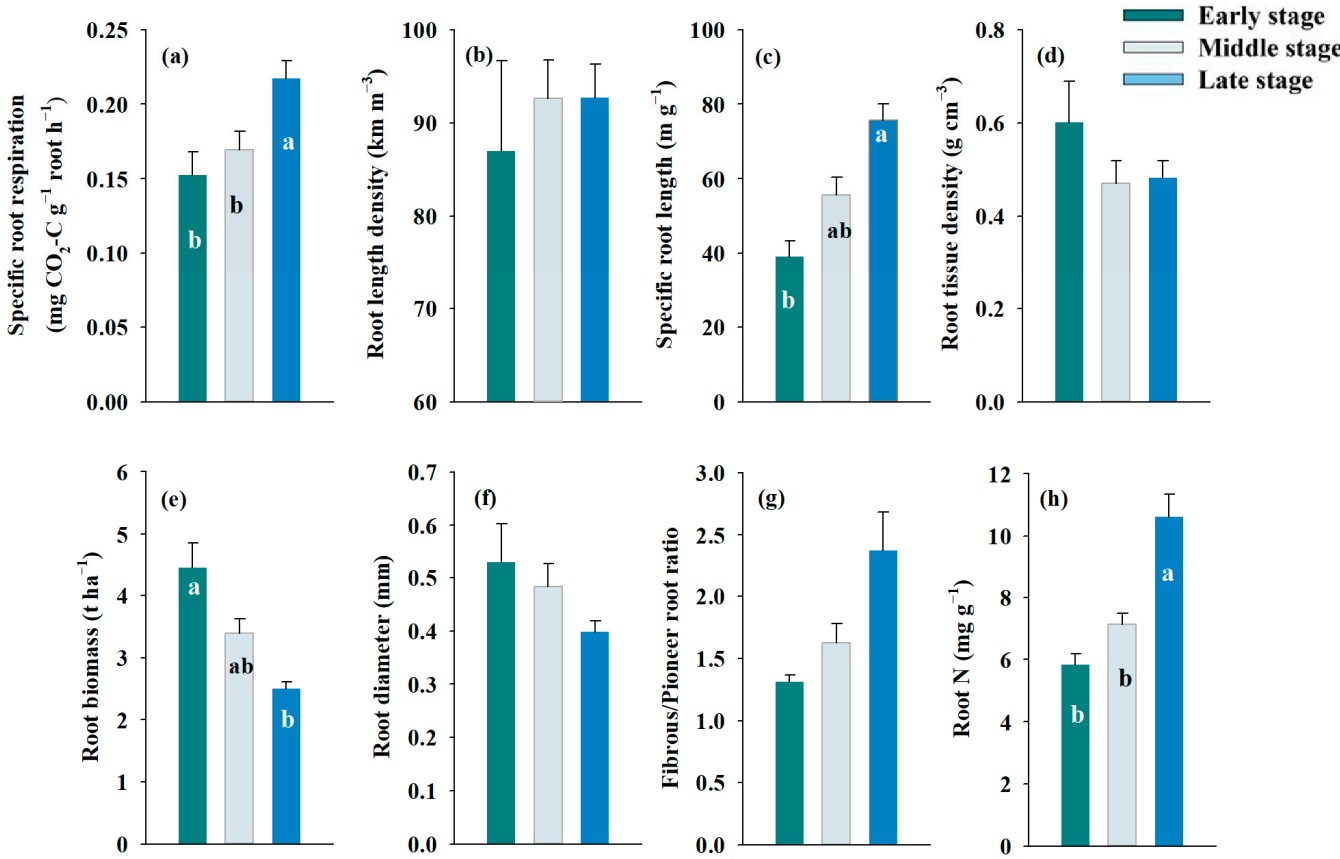

**Figure 4.** Fine-root traits in the top 20 cm of soils at three successional stages. Including specific root respiration (**a**), root length density (**b**), specific root length (**c**), root tissue density (**d**), root biomass (**e**), root diameter (**f**), fibrous/pioneer root ratio (**g**), and root N (**h**). Different lowercase letters indicate significant differences among successional stages, $p < 0.05$. Bar charts represent mean values, vertical bars represent the standard error.

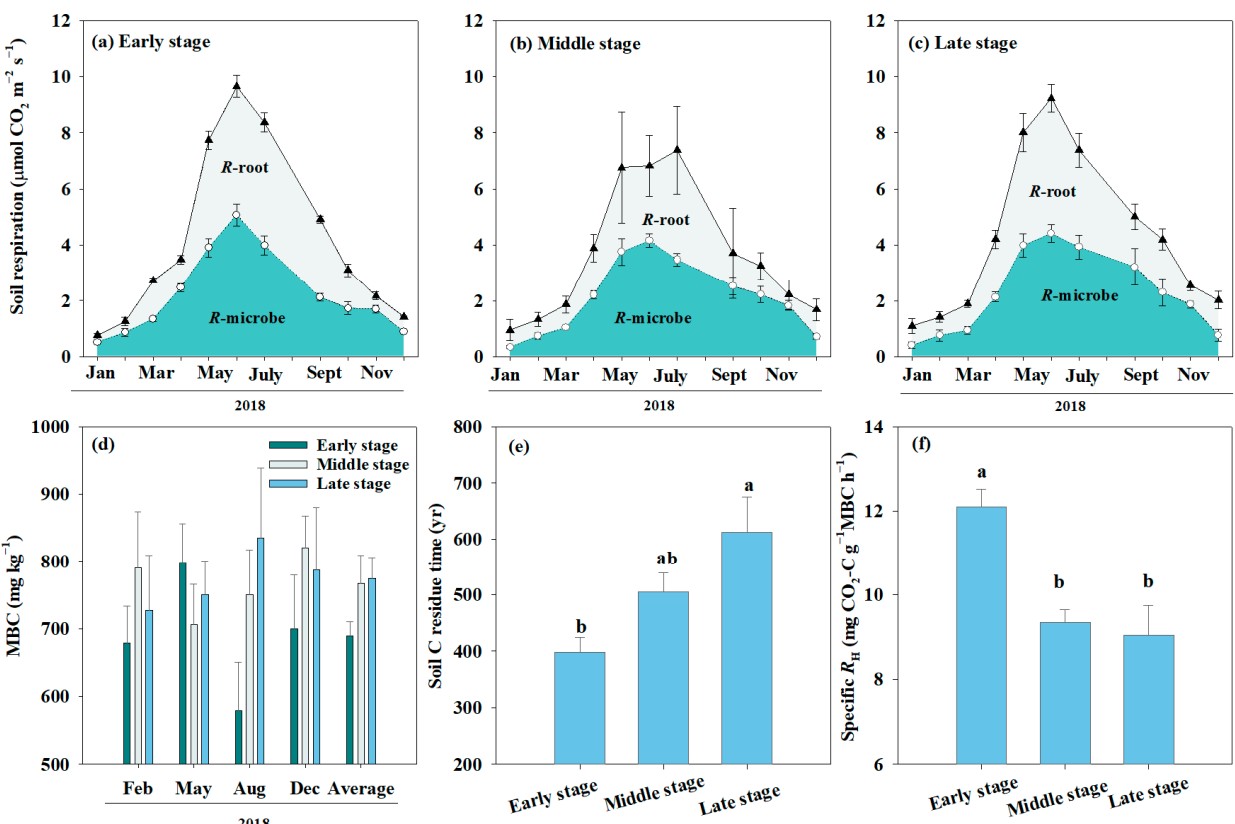

**Figure 5.** Soil respiration (panels (**a**–**c**)), microbial biomass carbon (panel (**d**)), specific microbial respiration (panel (**e**)), and soil carbon residence time (panel (**f**)) across three successional stages (Mean (SE). Closed circles and solid lines indicate soil $CO_2$ flux in Natural Control plots. Open circles and dashed lines denote soil $CO_2$ flux in root trenching plots. The area in green represents microbial respiration, whereas the area in light green represents root respiration. Abbreviations: MBC: microbial biomass carbon, *R*-root: root respiration, *R*-microbe: soil microbial respiration. Different lowercase letters indicate significant differences among successional stages, $p < 0.05$. Dots and bar charts represent mean values, vertical bars represent the standard error.

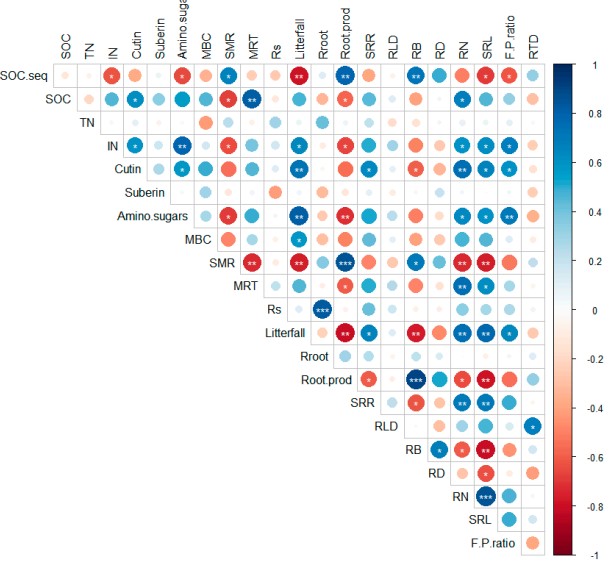

**Figure 6.** Relationships among SOC accumulation rate, fine-root functional traits, and soil properties. The size of the circle is proportional to the $R^2$ value. Blue dots indicate a positive correlation, while

red ones are negative. Abbreviations: SOC seq: SOC sequestration, IN: inorganic nitrogen, SRS: specific root respiration, MRT: mean carbon residence time, SMR: specific microbial respiration, $R$s: soil respiration, RLD: root length density, RMD: root mass density, Root N: root nitrogen concentration, SRL: special root length, F.P. root. ratio: fibrous/pioneer root ratio, RTD: root tissue density, $R_{root}$: root respiration, MBC: microbial biomass carbon. Stars represent statistical significance: * $p < 0.05$, ** $p < 0.01$, *** $p < 0.001$.

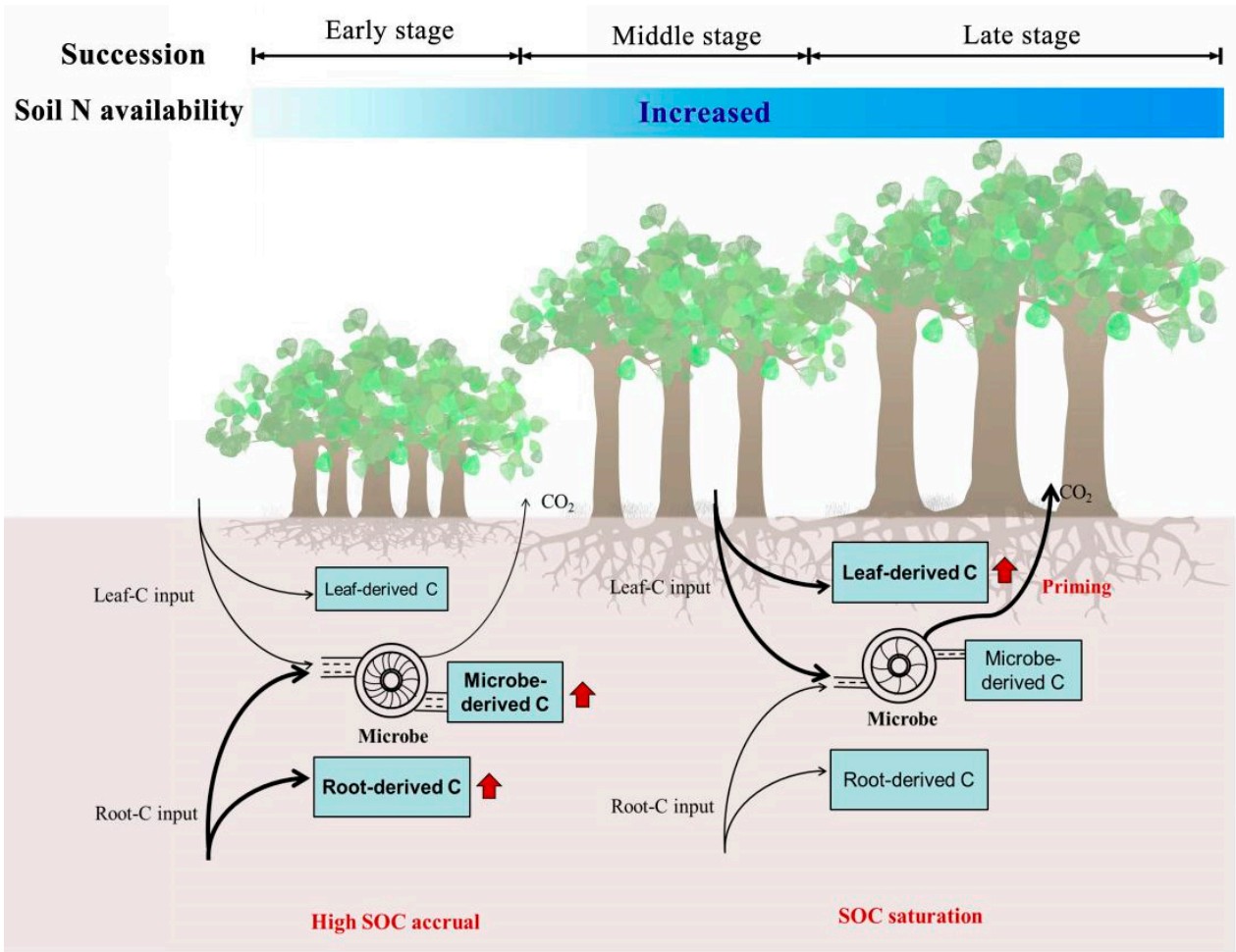

**Figure 7.** Improved conceptual framework explaining the process of SOC accrual in subtropical successional forests. The thickness of black arrows represents the size of soil C flux. Red arrows indicate the increasing C pool. The number of blades in the pump is positively correlated with microbial C turnover.

## 4. Discussion

### 4.1. Effects of Plant-Derived C Inputs on Soil Organic Carbon Accrual

Understanding the potential mechanisms controlling soil organic carbon (SOC) accrual across the forest succession is crucial for predicting long-term forest SOC dynamics [45,46]. In this study, we found that the SOC accrual rate significantly increased along the forest succession, with rapid soil C accumulation at the early-successional stage but a slight change at the mid- and late-successional stages (Figure 1). It is consistent with other studies on long-term forest chronosequences [46]. In the subtropics, high nitrogen (N) deposition in the young forests is coupled with "sufficient and synchronous water and heat availability", resulting in high net ecosystem productivity and rapid SOC accrual [47]. However, the limited SOC accrual in late-successional forests indicated that soil C saturation was likely occurring. According to the hypothesis of soil C saturation, the capacity of soils to store C is finite; additionally, soil C storage potential may control SOC sequestration [48,49]. The

saturation deficit generally decreased with the forest succession, potentially explaining the successional change in soil C accrual rates [50].

Litter and plant roots are the main resources of SOC in forest ecosystems, while their relative contributions to SOC accrual remain highly elusive [9,51]. In this study, we provided robust evidence that C inputs from plant roots drove SOC accrual more than those from above-ground litter in early-successional forests (Figure 2a). In contrast, C inputs from litterfall considerably contributed to SOC accrual from the mid- to late-successional stages (Figure 2a), although SOC accumulation is low. Previous studies demonstrated that SOC accrual rate is closely related to changes in plant community composition, root production and functional traits, litterfall biomass, and/or microbial activities [9,31], which largely differ across the forest successional stages [25,26,29,30]. In our study, we found positive correlation of the SOC accrual rate with fine-root production but negative correlation with annual litterfall (Figure 3). Root production decreased along forest succession, suggesting that early-successional forests might receive more root-derived C than mid- and late-successional ones (Figures 2a and 3a). Higher root production may be attributable to greater stem density in early- than late-successional forests in Eastern China [25,52]. By contrast, annual litter biomass increased along forest successional gradients, resulting in more leaf-derived C sequestered in the late- than early-successional forests (Figures 2a and 3b).

In addition to root biomass and production, SOC accrual was also correlated with root morphological and physiological traits (Figure 6). These root functional traits have been shown to vary across different stages of forest succession, which will likely result in the changes in below-ground energy use efficiency and the quality of root-derived residues [53]. For example, lower specific root respiration (SRR) in the early-successional forests resulted in a lower C cost per root biomass than those in the mid- and late-successional forests (Figure 4a), causing more root-derived compounds sequestered in the soil [54]. Likewise, root diameter tended to be thicker and root tissue was denser in the early- than late-successional forests (Figure 4) [25]. These traits usually indicate greater lignin concentration in roots, which might promote soil C stabilization [55,56].

In this study, we used two groups of widely-accepted biomarkers, (i.e., cutin and suberin) to quantify the relative contributions of aboveground leaf- and root-C to SOC accumulation. Due to methodological limitations, branch-derived-C cannot be determined from soil organic matter. Leaf litter accounts for more than 70% of the above-ground litterfall in the study area. It thus seems reasonable to explore the effects of above- and belowground carbon inputs on soil carbon accrual through analyzing cutin and suberin biomarkers, although the effects of branch-derived-C inputs were not examined.

*4.2. Microbial Controls on SOC Accrual*

The balance between microbial decomposition of organic C and the retention of soil microbial necromass governs SOC accrual [57]. In this study, we found that microbial anabolism, producing necromass C sequestrated in soils, drove the rapid SOC accrual in early-successional forests (Figure 2b). It is consistent with other findings and may be linked to changes in microbial C turnover and soil N availability [57]. Previous studies widely accepted that lower microbial C turnover could promote soil C accrual, since more C would be retained as soil organic matter or microbial biomass [58]. In contrast, our study provided evidence that faster microbial C turnover might stimulate greater microbial metabolite production in the early- than late-successional forests, contributing to soil C accrual (Figures 2b and 5f). Soil N availability, which is strongly related to microbial C turnover, may regulate the microbial C pump effect observed during forest succession. In early-successional forests, low soil N content might prime faster microbial C turnover (represented by specific microbial respiration) than those in mid- and late-successional forests (Figures 2 and 5). In contrast, microbes require less C to forage N from soil in a N-enriched condition in late-successional forests, potentially leading to the suppression in microbial C turnover and metabolite production (Figures 2 and 5).

Contrary to microbial anabolism that accelerated soil C accrual in early-successional forests, greater soil microbial biomass and enzyme activity potentially caused higher soil C loss in late-successional ones (Figure 5d, enzyme data shown in Liu et al., 2021 [26]). Meanwhile, leaf-derived C accumulation in soils caused only a slight increase in SOC stock in late-successional forests, suggesting that fresh C inputs induced the priming effects on old soil C decomposition (Figure 7) [59–61]. Soil organic carbon, microbial available carbon, and litter quality was usually greater in late- than early-successional forests, probably resulting in stronger microbial priming [26,62,63]. Overall, greater microbial biomass and soil priming effects largely explained the little changes in soil C stock over time in late-successional forests.

The contributions of fungal and bacterial necromass to soil C accrual reflect their roles in regulating SOC dynamics [30,64]. From the early- to mid-successional stages, fungal residues contributed more to soil C accrual than bacterial ones (Figure 2b), suggesting that SOC formation and stabilization was primarily influenced by the metabolic processes of the fungal community [65]. This is consistent with previous studies that fungal residues are more stabilized than bacterial ones [66,67]. In addition, the cell walls of soil fungi are primarily composed of chitin, which is more recalcitrant to decomposition than bacterial residues [68]. By contrast, from the mid- to late-successional stages, bacterial residues accumulated significantly, but fungal ones changed slightly (Figure 2b), which may be attributable to greater mineral N concentration at these stages than early-successional ones [66].

### 4.3. Implications for Future Experiments and Forest Management

Understanding the effects of forest succession on soil C accrual and revealing the associated mechanisms in forest ecosystems are essential to mitigate global change. Our results provide insights to explain SOC dynamics across the succession in subtropical forests, which may offer some suggestions for future experiments and forest management. First, we found that root production and microbial C turnover drove rapid SOC accrual in early-successional subtropical forests. However, the underlying mechanisms regulating belowground C input and soil physical-chemical protection remain limited, which should be further explored in the future studies. Our study also highlights the importance of the subtropical soil C sink in the current global C cycle. Great attention should be given to future forest management and conservation applications in subtropical regions.

Second, bacteria and fungi contribute differentially to the soil C pool [66]. In this study, fungal residues contributed more to soil C accrual than did bacterial ones from the early- to mid-successional forests (Figure 2b), exhibiting the importance of soil fungi in soil C cycling. Soil fungi include mycorrhizal and saprotrophic hyphae; it is difficult to distinguish their relative contributions to soil C accrual. A better understanding of these contributions deserves to be explored via $^{13}C$ isotope analysis in the future. Third, plant-derived C inputs and microbial C transformation play different roles in SOC accrual across the forest succession. The soil priming effect may be an important underlying mechanism driving soil C saturation in late-successional forests. This effect needs to be further studied with new techniques and approaches. Moreover, the differential roles of plant-derived C inputs and microbial activity to SOC accrual across different successional stages should be considered in current land-surface models to forecast long-term forest soil C dynamics.

### 5. Conclusions

Understanding how plants and microbes influence soil C formation and sequestration is crucial to predict SOC dynamics in the future climate. Based on a long-term observation (2004–2018) in subtropical forests, we found that SOC accumulated rapidly in early-successional forests, but changed slightly at the mid- and late-successional stages. Rapid SOC accrual in early-successional forests is likely a result of great root production and microbial C turnover. In contrast, leaf-derived C inputs may cause priming effects on old soil C decomposition in late-successional forests, probably causing little change in the

soil C pool. Our study highlights the importance of root production and microbe-derived C inputs in SOC accrual in early-successional subtropical forests, which could be incorporated into Earth system models to improve model performance and the feedback of the forest C cycle to climate change.

**Supplementary Materials:** The following supporting information can be downloaded at: https://www.mdpi.com/article/10.3390/f13122130/s1, Table S1: Stand characteristics at different stages of forest succession (Mean(SD)).

**Author Contributions:** Conceptualization, R.L. and X.Z.; Investigation, R.L., Y.H., Z.D. and G.Z.; Supervision, X.Z.; Writing—original draft, R.L.; Writing—review and editing, L.Z., X.W., N.L., E.Y., X.F., C.L. and X.Z. All authors have read and agreed to the published version of the manuscript.

**Funding:** This research was financially supported by the National Natural Science Foundation of China (Grant No. 32001135, 31930072, 31901200, 32071593), the Fundamental Research Funds for the Central Universities (2572022BA06), and the Natural Science Foundation of Heilongjiang Province of China (ZD2021C002).

**Acknowledgments:** We would like to thank Murphy Stephen at Yale University for his assistance with English language and grammatical editing of the manuscript.

**Conflicts of Interest:** The authors declare no conflict of interest.

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
