# Peer review of "Root Production and Microbe-Derived Carbon Inputs Jointly Drive Rapid Soil Carbon Accumulation at the Early Stages of Forest Succession"

_forests, doi:10.3390/f13122130_

Round 1
Reviewer 1 Report (Previous Reviewer 3)
I think the paper should be accepted.
Author Response
Thank you for your comments.
Reviewer 2 Report (New Reviewer)
This study focuses on the evolution of organic carbon inputs to soil at different stages of succession. There are some questions and comments that need clarification.
1. Why have the authors restricted themselves to studying only fine roots? And only 0–20 cm depth?
2. “Soils were sampled using PVC cylinders. . .” PVC cylinders? Often, the problem is taking samples with steel cylinders, even at such a small depth. High soil density, rocks and coarse roots make it impossible to sample with steel cylinders or drills. How could the authors use PVC cylinders?
3. You have measured the annual litterfall. Furthermore, carbon in the litterfall was defined as leaf-derived C. But litterfall is composed not only of leaves but also of dead and fallen branches. Why have the authors not divided the leaf-derived C and branches-derived C?
Author Response
This study focuses on the evolution of organic carbon inputs to soil at different stages of succession. There are some questions and comments that need clarification.
- Why have the authors restricted themselves to studying only fine roots? And only 0–20 cm depth?
[Response] Thanks for the comments. Fine roots play a key role in soil carbon sequestration through mediating below-ground carbon inputs and soil organic matter decomposition. In this study, we explored the effects of fine-root production and functional traits on soil carbon accrual by analyzing their correlation with soil carbon accumulation rate. Due to technical limitations, it is hard to measure the production of coarse roots. Hence, we did not examine the effects of coarse root growth on soil carbon dynamics. Topsoils (0-20 cm) are the most active zone of biogeochemical cycle. We sampled the soils and fine roots at the depth of 0-20 cm to examined plant and microbial controls on soil carbon sequestration in the top soil layer. We did not quantify these effects in the deeper soil layers in this study.
- “Soils were sampled using PVC cylinders. . .” PVC cylinders? Often, the problem is taking samples with steel cylinders, even at such a small depth. High soil density, rocks and coarse roots make it impossible to sample with steel cylinders or drills. How could the authors use PVC cylinders?
[Response] Sorry for this confusion. In 2004, soils were sampled using PVC cylinders. In order to reduce the methodological errors of soil sampling, we continued to use PVC cylinders to sample soils and roots in 2013 and 2018. PVC cylinders are hammered into the soil and then dug out with a spade. These PVC cylinders were then sent to the laboratory for further processing and analysis. We have added the related information in Line 135-136 of revised manuscript.
- You have measured the annual litterfall. Furthermore, carbon in the litterfall was defined as leaf-derived C. But litterfall is composed not only of leaves but also of dead and fallen branches. Why have the authors not divided the leaf-derived C and branches-derived C?
[Response] Thanks for the comments. In this study, two groups of widely-accepted biomarkers, (i.e., cutin and suberin) were used to quantify the relative contributions of aboveground leaf-, root-C to SOC accumulation. Due to methodological limitation, we cannot divide branches-derived C from soil organic matter. While leaf litter accounts for more than 70% of the above-ground litterfall in the study area. It thus seem reasonable to explore the effects of above- and belowground carbon inputs to soil carbon accrual through analyzing cutin and suberin biomarkers, although the effects of branches-derived C inputs (account for 15%-20% of aboveground litterfall) were not examined. We have added the related information in Lines 458-465 of revised manuscript.
Reviewer 3 Report (New Reviewer)
The paper entitled “Root production and microbe-derived carbon inputs jointly drive rapid soil carbon accumulation at the early stages of forest succession” investigated plant and microbial contribution to soil organic carbon in successional subtropical forests. Leaf, root and microbial biomarkers, root and leaf litter inputs, and microbial C decomposition were measured in this study. The paper needs major revision. The most important thing is that in discussion section, the authors repeat the results again and they have not discussed the results and compare those with other studies. What was the reason of your result in agreement and disagreement of other studies? The authors must strengthen the discussion. Some detailed comments as follow:
1. It is better to insert line numbers to address the comments easily.
2. Section 2.1: Please provide a map of study area.
3. Section 2.4: How did the authors sample roots?
4. Did you use 15% alcohol solution on the roots, to prevent mould and microbial degradation?
5. What did you measure as root morphology?
6. How do you measure root length by WinRHIZO?
7. Some valuable and relevant papers that can help you describe section 2.4 are:
https://doi.org/10.1007/s11104-021-05231-1
https://doi.org/10.1007/s40333-018-0021-2
https://doi.org/10.1016/j.foreco.2020.118873
https://doi.org/10.1016/j.catena.2022.106410
https://doi.org/10.1016/j.ecoleng.2011.03.026
https://doi.org/10.1016/j.ecoleng.2021.106309
https://doi.org/10.1007/s11104-022-05764-z
https://doi.org/10.1016/j.foreco.2018.02.031
https://doi.org/10.3390/f10040341
https://doi.org/10.1002/2015JF003632
8. Section 2.5: How much the dimension of the trenches?
9. Figure 1b: modify the axes of the figure.
10. Figure 3c and 3d: Modify the figures with equations.
11. Figure 4: Are the figures show mean values? If yes, mention it in text or figure.
12. Figure 4g: it looks that there is a significant difference among different stages. Check your results again.
13. In all figures that you did a comparison it is better to add a sentence likeà different lower cases show significant difference.
Author Response
Thank you for your comments. Please see our responses in the attachment.

Round 2
Reviewer 3 Report (New Reviewer)
Good job!
Author Response
Thank you very much

This manuscript is a resubmission of an earlier submission. The following is a list of the peer review reports and author responses from that submission.
Round 1
Reviewer 1 Report
Dear Authors,The methodology of sample preparation and analysis is not presented appropriately.
The results are not according to the sampling years.
Other suggestions are mentioned in the attached reviewed manuscript.
Regards,

Reviewer 2 Report
Dear authors,
I have some suggestions to your work.
Line 23: Please, remove the bold of the word “Plants”.
Lines 107-108: I wonder if the sentence “Soils are classified as hilly red and yellow earths, and pH ranges between 4.5-5.1 (Acrisols and Cambisols in the FAO soil classification, respectively) [34, 35]” can be reordered in the following way: “Soils are classified as hilly red and yellow earths (Acrisols and Cambisols in the FAO soil classification), and pH ranges between 4.5-5.1 [34, 35]”.
Line 112: “Castanopsis fargesii community [25, 26]).”à Suggestion: “Castanopsis fargesii community) [25, 26].”
Lines 112-113: Please, check if the following sentence “The slope and elevation of three successional forest stands …” can be written as “The slope and elevation of THE three successional forest stands …”
Line 118: “Liu et al., (2019)”à “Liu et al. (2019)”. The same in line 172: “Shi et al., (2005)”à “Shi et al. (2005)”. The same in line 217:” Brookes et al., (1985)”
Line 123: “Soil Samples”à”Soil samples”
Line 136: “Three solvent chemical extracts”à ”The three solvent chemical extracts”
Line 147: “Zhang &”à”Zhang and”
Line 176: “2018, respectively”à The word “respectively” is not needed.
Line 179: “Roots were then”à The word “then” is not needed.
Line 213: “Soil microbial biomass C”à Please, mention if you sieved the samples and if the soil for the assay was fresh soil.
Lines 217-218: “Samples were mixed with 20 ml K2SO4 (0.5M) and distilled using a shaker for 30 minutes”àI would write “Samples were mixed with 20 ml K2SO4 (0.5M) and shaked for 30 minutes”.
* Please, review Fig. 1b. It should be SOC stock, in the y axe is written SOC concentration.
Line 243: “with an average rate of 1.7 Mg C ha−1 yr−1 (Fig.1 b)”. I wonder why you write Mg C in the text and ton in the Figure.
Line 247: Perhaps in Figure 1 or in the caption of this figure can be written the years: 2004, 2014 and 2018.
Line 250: “The insets in panel (a and b) showed the SOC”à” The insets in panel (a and b) SHOW the SOC”
Line 276: “insect”à”INSET”. See also Line 182.
Line 277: In Figure 2 b “Total neromass”à”Total NECROMASS”
Line 323: Add ***: P<0.001
Line 326: Please, add CO2 in “umol m-2 s-1”
Line 374: “In our study, we found positive correlation of SOC accrual rate with fine-root production but negative correlation with annual litterfall”. Are there other authors that have found this negative correlation?
Line 411: “(Fig. 5 d, enzyme data shown in Liu et al., 2021 [26]).”à ”(Fig. 5 d, enzyme data shown in Liu et al. (2021) [26]).”
Lines 448-450: “However, we did not carefully investigate this effect in this study, which needs to be further studied with new techniques and approaches”à Suggestion: “This effect needs to be further studied with new techniques and approaches”
Reviewer 3 Report
Root production and microbe-derived carbon inputs jointly drive rapid soil carbon accumulation at the early stages of forest succession
This paper aim to study the importance of root production and microbial anabolism in SOC accrual at the early stages of forest succession. The long-term monitoring results showed that SOC accumulated rapidly at the early-successional stage, but changed little at the mid- and late-successional stages. Incorporating these effects of belowground C inputs on SOC formation and accumulation into Earth system models might improve model performance and projection of long-term soil C dynamics. I think the paper should be accepted with a few revisions. The detailed suggestions are as follows:
1. Line 87, change “in the main driver”to “the main driver”
2. Line 430, change“Understanding effects”to “Understanding the effects”